# Adsorption of Hexavalent Chromium by Sodium Alginate Fiber Biochar Loaded with Lanthanum

**DOI:** 10.3390/ma14092224

**Published:** 2021-04-26

**Authors:** Xinzhe Sun, Peng Guo, Yuanyuan Sun, Yuqian Cui

**Affiliations:** 1College of Environmental Science and Engineering, Qingdao University, Qingdao 266071, China; 2018025497@qdu.edu.cn; 2School of Chemical Engineering, China University of Petroleum (East China), Qingdao 266580, China; guopeng@upc.edu.cn

**Keywords:** wet spinning, lanthanum oxide, biochar, chromate, adsorption

## Abstract

Lanthanun oxide (La_2_O_3_) is a lanthanum chemical compound incorporates a sensible anionic complexing ability; however, it lacks stability at a low pH scale. Biochar fibers will give the benefit of their massive space and plethoric uses on the surface to support a metal chemical compound. Herein, wet spinning technology was used to load La^3+^ onto sodium alginate fiber, and to convert La^3+^ into La_2_O_3_ through carbonization. The La_2_O_3_-modified biochar (La-BC) fiber was characterized by SEM, XRD and XPS, etc. An adsorption experiment proved that La-BC showed an excellent adsorption capacity for chromates, and its saturation adsorption capacity was about 104.9 mg/g. The information suggested that the adsorption was in step with both the Langmuir and Freundlich models, following pseudo-second-order surface assimilation mechanics, which showed that the Cr (VI) adsorption was characterized by single-phase and polyphase adsorption, mainly chemical adsorption. The thermodynamic parameters proved that the adsorption process was spontaneous and endothermic. The mechanistic investigation revealed that the mechanism of the adsorption of Cr (VI) by La-BC may include electrostatic interaction, ligand exchange, or complexation. Moreover, the co-existing anions and regeneration experiments proved that the La-BC is recyclable and has good prospects in the field of chrome-containing wastewater removal.

## 1. Introduction

Chromium (Cr) is amongst the foremost typical and venomous significant metal ions, which is widely found in wastewater from a variety of industries, such as textiles, metallurgy, metal electroplating, and tanneries [1]. Chromium can cause serious environmental problems and harm human health due to its problematic characteristics of bioaccumulation, non-biodegradability and potential carcinogenicity in the food chain [2,3].

Generally speaking, chromium in nature mainly exists in two stable states in a wide pH range: trivalent chromium (Cr (III)) and hexavalent chromium (Cr (VI)). Cr (III) is non-toxic and can be effectively removed by precipitation (such as chromium hydroxide) or adsorption [4,5,6,7]. Cr (VI) are more dangerous because it has toxicity, and higher solubility and mobility [4]. It appears in the forms of chromate (CrO_4_^2−^, HCrO_4_^−^) and dichromate (Cr_2_O_7_^2−^), which have caused great environmental concern [3,8]. As a priority toxic pollutant identified by the US environmental protection agency, Cr (VI) can cause serious toxic and detrimental effects to human health and ecosystems. When the Cr (VI) content in drinking water exceeds 0.05 mg/L and that in water for different uses exceeds 0.1 mg/L, it can cause harm to human health, such as kidney, liver and stomach damage, skin allergies and lung cancer [9], but in contaminated natural water and industrial wastewater, Cr (VI) tends to be more than 200 mg/L [10,11]. Therefore, practical methods must be developed to remove Cr (VI), in order to protect the aquatic environment.

Compared with strategies such as membrane separation, biodegradation, electrochemical oxidation, solvent extraction and other treatment methods, the capture of Cr (VI) using adsorbing media has numerous advantages, such as being a non-toxic and harmless process, easy operation, versatility, and low cost [12,13,14]. The adsorbents with different sites will interact with Cr (VI) through different mechanisms, including direct precipitation, electrostatic interaction, ligand exchange, intra-sphere complexation and redox [1,15,16,17]. Many of the naturally available adsorbents such as sawdust, pine needles and seaweed have the disadvantages of a low adsorption capacity and a slow adsorption process. Thus, in order to realize the unification of economic and environmental benefits, there is a need to create innovative and economical adsorbents for composite materials with high efficiency [18].

Biochar (BC) has recently received significant attention because of its beneficial surface physical/chemical properties for the removal of heavy metals from wastewater [19,20,21]. However, the surface of the original BC is usually negatively charged, with a low anion removal ability. Rare earth metal oxides can participate in chemical reactions as active components, and have been widely used in petroleum, metallurgy, ceramics, textiles and other fields. Lanthanum oxide (La_2_O_3_) has a good anionic complexation ability, but lacks stability under low pH conditions. In actuality, biochar fibers can take advantage of their large surface area and abundant surface functional groups to support lanthanum oxide; meanwhile, the pores in biochar could enhance the dispersion of La and improve the consumption efficiency, so biochar is a viable option to support La incorporation [22]. Recently, biochar materials with lanthanum oxide dropped have been extensively studied, and they exhibited several advantages in removing oxygen-containing acids, such as strong adsorption selectivity, remarkable adsorption capacity, a high removal rate and a wide pH range. According to Yang Bei (2019), through inner-sphere complexation, La-dropped biochar can form an La-O-P chemical bond with phosphate to remove PO_4_^3−^ in sewage [23], while Haiyang Yang (2019) stated that Cr (VI) adsorption might be related to the outer-sphere complexation. Therefore, we speculate that chromate can also bond with La-BC with the same chemical bond [24,25]. In addition, other literature data shows that lanthanum-containing materials have an excellent effect on the removal of phosphate, antimonate and arsenate in water [14,21,26,27,28,29,30,31]. As a great biochar doping component, La can improve the adsorption capacity of anions. However, there are few reports in the literature on the use of La-doped BC to remove Cr (VI). Moreover, the adsorption capacity of Cr (VI) by La-based adsorbents is also low in these relevant literatures. For example, the adsorption capacity of hexavalent chromium in chitosan and lanthanum mixed oxyhydroxide is only 48.3 mg/g [32]. In order to investigate the adsorption capacity of La for chromate, a lanthanum-oxide–loaded carbon fiber material was prepared by calcination at 90 0 °C. By using La-BC, Cr (VI) can be effectively removed. The possible mechanism is discussed in detail. It is proved that the adsorption effect of La-BC on Cr (VI) is better than that of other La-based adsorbents reported in the literature. Beyond that, La-BC is recyclable, and has good prospects in the field of chrome-containing wastewater removal.

## 2. Materials and Methods

### 2.1. Materials

The reagents included lanthanum chloride, sodium alginate (alighting), potassium dichromate, hydrochloric acid, sodium hydroxide, sulfuric acid, phosphoric acid, sodium chloride, sodium sulfate, sodium nitrate, sodium bicarbonate, disodium phosphate, acetone, and diphenylcarbazide. All of the chemicals and reagents were analytically pure, and were produced by the Sinopharmaceutical chemical reagent co., Ltd., Shanghai, China except for the sodium alginate (AR, Shanghai Aladdin Bio-Chem Technology Co., Ltd., Shanghai, China). All of the solutions were prepared using deionized water.

### 2.2. Preparation of the Adsorbent

The La^3+^-loaded sodium alginate fiber (La-SA) was prepared by wet spinning with 4% sodium alginate solution as the spinning solution and lanthanum chloride (LaCl_3_) solution with the same concentration as the coagulate bath. The La-SA was placed in an oven at 30 °C and dried for 24 h. The dried La-SA was transferred to a tube furnace and pyrolyzed at 900 °C for 1 h (with a heating rate of 5 °C/min). After being ground and sifted, the biochar (La-BC) loaded with lanthanum oxide (La_2_O_3_) was obtained for the experiment. Under the same concentration of sodium alginate, pure sodium alginate fibers were prepared with a mixture of 5% NaCl and 4% HCl as the coagulation bath. Through the same pyrolysis conditions, pure sodium alginate biochar (SA-BC) and pure lanthanum oxide (La_2_O_3_) were prepared in order to compare the adsorption performance of the samples.

### 2.3. Characterization

The Brunauer-Emmett-Teller equation was used to determine the specific surface area, and the pore structures were calculated by nitrogen adsorption/desorption isotherms carried out on a surface area and porosity analyzer (3H-2000PS2, Best Instrument Technology Co., Ltd., Beijing, China). The surface morphology of material was observed by scanning electron microscopy (SEM, SU8020, Hitachi Hi-Tech Co. Ltd., Tokyo, Japan). Besides this, the phase structure of the materials was examined by X-ray diffraction (XRD, DX2700, Shanghai Precision Instrument Co., Ltd., Shanghai, China) at a scan rate (2θ) of 6°/min, from 10° to 80°, and at an accelerating voltage of 40 kV. X-Ray photoelectron spectroscopy (XPS, ULTRAESCALAB 250Xi, Thermo Fisher Scientific, Waltham, MA, USA) was used to analyze the surface functional groups and the valence state of the chromium with monochromatic 150 WAl Kα radiation.

### 2.4. Adsorption and Desorption Experiments

A series of adsorption and desorption experiments were carried out in order to evaluate the adsorption properties of the La-BC and analyze its adsorption mechanism. Potassium dichromate (2.827 g) was dissolved in distilled water (1000 mL) to make a 1000 mg/L dichromate stock. Then, the stock was diluted to produce dichromate solutions of different concentrations. The effects of the carbonization temperature (700–1000 °C), solution pH (2–10), adsorption isotherm (initial concentration 20–1000 mg/L), adsorption kinetics (shaken for 3 h in oscillator), adsorption thermodynamics (adsorbent temperature 298–318 K) and co-existing acid ions on the adsorption of Cr (VI) were analyzed in turn. In order to assess the reusability of La-BC as a Cr (VI) adsorbent, five adsorption/desorption cycles were conducted. The filtrate was filtered with a 0.45 μm nylon membrane syringe after adsorption. The clear solution was extracted, and the possible leaching concentrations of Cr^6+^ and La^3+^ in the adsorbed solution were determined by a visible light spectrophotometer and an inductively coupled plasma emission spectrometer, respectively.

In all of the adsorption experiments, the dosage of La-BC was 1 g/L. The pH of all of the solutions was not adjusted separately, except for the pH experiment. Similarly, all of the experiments were conducted at room temperature (298 K or 303 K), except the adsorption thermodynamics experiment. Due to the indoor temperature difference in different seasons, the specific experimental conditions are described in detail in the results and discussion sections below. In addition, in all of the charts in this paper, the data—such as the Cr (VI) solution concentration and the adsorption capacity—were calculated by the total Cr (VI) content.

## 3. Results and Discussion

### 3.1. Preliminary Experiments

Before the formal batch adsorption experiment, the adsorption properties of pure sodium alginate biochar (SA-BC), pure lanthanum oxide (La_2_O_3_) and La-BC were compared through the preliminary experiment under the conditions of pH 4.6, 300 rpm, 303 K, contact time 5 h and adsorbent dosage 1 g/L (Figure 1). The data showed that the Cr(VI) adsorption capacity of the sample was ranked as La-BC > La_2_O_3_ > SA-BC, which may be due to the following reasons: (1) the surface of SA-BC is negatively charged, which leads to its low ability to remove anionic groups such as Cr_2_O_7_^2−^ and CrO_4_^2−^, and the main adsorption process is physical adsorption caused by its porous structure; (2) La_2_O_3_ has a good anion complexation ability, but it is not stable in an acidic solution. Therefore, loading La_2_O_3_ onto SA-BC to prepare La-BC is a win–win situation. It can take advantage of the larger surface area and rich surface functional groups of biochar to support La_2_O_3_, and the pores in biochar can enhance the dispersion of La, improve the stability of La_2_O_3_, and aid the removal efficiency of Cr (VI).

### 3.2. Characterization of the La-BC

Appendix A shows some of the physical properties of carbonized adsorbents at different temperatures. According to the nitrogen adsorption–desorption curve (Appendix A), it can be found that all of the four carbon adsorbents are type IV with H4 hysteresis loops, indicating that the lanthanum-supported biochar, through high temperature carbonization, is a mesoporous material [28,33]. The specific surface area and porosity of the adsorbent have an important influence on its adsorption capacity [20,34]. Generally speaking, the larger the specific surface area is, the more adsorption points are available, and the larger the adsorption capacity will be. The specific surface area of La-BC (900 °C) reached 177.41 m^2^/g, while the specific surface areas of La-BC (700 °C), La-BC (800 °C) and La-BC (1000 °C) were 8.21 m^2^/g, 38.61 m^2^/g and 52.08 m^2^/g respectively, indicating that La-BC (900 °) might be the adsorbent with the highest adsorption capacity. In addition, by comparing the pore volume, it was found that the pore volume of La-BC (900 °C) was 0.2515 cm^3^/g, which was larger than the other three adsorbents. Larger pore volumes also facilitate the adsorption of the target elements.

SEM images, TEM images and element mapping were used to research and analyze the surface morphology and element distribution of La-BC (900 °C) (Figure 2). As a porous material, biochar is a good substrate for lanthanum deposition. The sodium alginate biochar supported by lanthanum was fibrous, with a diameter of about 20–50 μm. The surface was rough and irregular, with a large number of acicular crystals and loose pores distributed upon it (Figure 2a–c). The same morphology was also shown in TEM images, in which the fractured biochar fibers were supported by needle-like or rod-like crystals (Figure 2d–f). The lattice fringe of La-BC can be observed more clearly through the analysis of the high-resolution TEM image (Figure 2g). Compared with the standard card La_2_O_3_ (PDF#05-0602), the lattice spacings of 0.335 nm and 0.346 nm both correspond to the (100) plane of La_2_O_3_. In the SEM-EDS element mapping image of the La-BC (900 °C), the La element can be observed evenly (Figure 2h), proving that the La element has been successfully loaded onto the surface of the biochar.

The XRD patterns of La-BC (900 °C) at four different carbonization temperatures are shown in Figure 3. The composition of its elements and substances was analyzed. It was found that the existing forms of La change with the increase of the calcining temperature. For La-BC at 700 °C and La-BC at 800 °C, there were diffraction peaks of La_2_CO_5_ (PDF#23-0320), while for La-BC at 900 °C and La-BC at 1000 °C, there were characteristic peaks of La_2_O_3_ (PDF#05-0602), which proves that La was successfully loaded onto the adsorption materials [30], and that the lanthanum-containing products were different at different carbonization temperatures.

In order to further identify the functional groups on the La-BC and to obtain the possible adsorption mechanism of Cr (VI), the sample La-BC (900 °C) was analyzed by X-ray photoelectron spectroscopy (Figure 4). All of the peaks are calibrated by using C 1s (284.3 eV) as a reference. The wide-scan XPS spectrum of La-BC showed that the elements C (63.4%) and O (28.96%) were the main components (Figure 4a). The peaks at the binding energies of 835.6 eV, 839.0 eV, 852.6 eV and 855.8 eV confirmed the introduction of La^3+^ on the biochar surface (Figure 4b).

### 3.3. Adsorption Isotherm

The adsorption isotherm refers to the relationship curve between the adsorption amount of the adsorbent and the equilibrium concentration of the adsorbent after the adsorption process occurs at the set adsorption temperature. This process can be fitted by the adsorption isotherm model (Appendix A). The correlation between the Langmuir and Freundlich isotherm models and the experimental data with different initial chromium concentrations was studied. The isothermal adsorption model fitting of carbonized La-BC at four temperatures is shown in Figure 5. Under the conditions of pH 4.6, 300 rpm, 303 K, contact time 5 h and a 1 g/L dosage of the adsorbent, adsorption experiments were carried out with four kinds of La-BC, in which the range of the initial potassium dichromate concentration was from 20 to 1000 mg/L. It was found that the adsorption capacity of La-BC (700–1000 °C) increased from 2.2 to 37.6 mg/g, 9.6 to 56.3 mg/g, 5.8 to 83.5 mg/g, and 3.2 to 39.6 mg/g, respectively. The increase of the initial concentration of the potassium dichromate solution might result in the improvement of the collisions between La-BC and Cr (VI), and might increase the driving force of removing Cr (VI), thus promoting adsorption [35]. The adsorption isotherm shows that, in the carbonized La-BC at four different temperatures, the adsorption capacity of 900 °C La-BC is higher, which may be attributed to the following two reasons: (1) as the specific surface area and pore size of 900 °C La-BC are larger than those of the other three kinds of La-modified biochar (Appendix A), there are more available adsorption sites, enhancing the adsorption capacity of 900 °C La-BC; (2) according to the XRD pattern (Figure 3), the lanthanum compound loaded with 900 °C La-BC and 1000 °C La-BC was La_2_O_3_, which could be used as an active adsorption site for metal ions. The Langmuir isotherm model showed that the saturated adsorption amounts of Cr (VI) with four kinds of La-BCs, respectively, were 41.4 mg/g, 67.4 mg/g, 104.9 mg/g and 49.9 mg/g (Appendix A). The fitting results showed that the adsorption process had consistency with both the Langmuir model and the Freundlich model, but conformed to the Langmuir model more, suggesting that the adsorption process should exist in monolayer and multilayer adsorption, especially the adsorption of a single molecule layer [35,36].

### 3.4. Adsorption Kinetics

Adsorption is a continuous dynamic process characterized by ‘external diffusion —internal diffusion’. This process can be fitted by the adsorption kinetic model (Appendix A). In order to evaluate the chromate adsorption rates of La-BC, the time-dependent sorption was conducted under the conditions of an initial Cr (VI) concentration of 200 mg/L, pH 4.6, 300 rpm, 303 K, contact time 5 h and a 1 g/L adsorbent dose. The relationship between the adsorption amount of Cr (VI) and the time of the La-BC carbonized at four temperatures is shown in Figure 6a. The adsorption process can be divided into three stages: a rapid adsorption stage, a slow adsorption stage, and an adsorption equilibrium stage. In the first 20 min, the adsorption reaction of Cr (VI) with La-BC was very rapid. The Cr (VI) adsorption capacities of La-BC (700–1000 °C) were 14.1 mg/g, 19.0 mg/g, 30.3 mg/g and 15.5 mg/g, which occupied the corresponding equilibrium adsorption capacities of 95.5%, 93.1%, 93.5% and 95.7%, respectively, and the adsorption capacity of La-BC at 900 °C was far higher than that of the other three adsorbents. The following 70 min is the slow adsorption phase. After 90min, there was no significant difference between the concentration of Cr (VI) and that after 3h in the solvent, showing that the adsorption process entered the equilibrium stage due to the saturation of the sites [1,37]. The differences between these stages can be explained as follows: (1) at first, the concentration of Cr (VI) at the interface of the adsorbent was the highest, forming a large adsorption dynamic gradient, which drove the Cr (VI) to occupy the adsorption site on the outer surface of the La-BC rapidly; (2) when Cr (VI) entered the interior of La-BC, the adsorption rate decreased due to the blockage of the pores and the decrease of the active adsorption sites; (3) when the slow adsorption continued, the adsorption rate gradually decreased until it reached the dynamic adsorption equilibrium.

In order to determine the adsorption mechanism, the equilibrium data were used to fit the pseudo-first-order and pseudo-second-order models (Appendix A). The fitting results showed that the pseudo second-order kinetic equation can better describe the adsorption behavior. In other words, the adsorption of Cr (VI) on La-BC was closer to chemisorption, which is carried out through polar functional groups sharing or exchanging electrons between the adsorbate and adsorbent [38,39].

### 3.5. Adsorption Thermodynamics

Temperature is an important factor affecting the adsorption effect. Temperature changes affect the value of the thermodynamic parameters (Appendix A). Under the conditions of an initial Cr (VI) concentration of 100–900 mg/L, contact time 5 h, pH = 4.6, adsorbent 300 rpm and a 1 g/L dose, the relationship between the adsorption amount of Cr (VI) with La-BC (900 °C) and the adsorption temperature is shown in Figure 6b. The increase of the adsorption temperature makes the adsorption amount increase. The calculated adsorption reaction enthalpy change ΔH was 41.66 KJ/mol, ΔS was 144.6 J/(mol·K), and the Gibbs free energy change ΔG < 0 at each temperature (Appendix A), which showed that the adsorption was a spontaneous endothermic reaction; in other words, the rising of the temperature lead to a prompt reaction [12,16,40].

### 3.6. Effect of the Solution pH on the Adsorption Experiment

Under the conditions of an initial concentration of 250 mg/L, 300 rpm, 298 K, contact time 5 h and an adsorbent of 1 g/L, the effect of the pH on the adsorption capacity of Cr (VI) with La-BC (900 °C) is shown in Figure 7a, and the comparison of the initial and final pH of the solution before and after adsorption is also demonstrated. The selected pH range was 2–10, within which the adsorption capacity of the chromate decreased significantly with the increase of the pH, and the maximum adsorption capacity was 68.52 mg/g, appearing at pH = 2. The existent form of chromate is related to the pH of the solution. At an acidic pH, the main types of chromate existent forms are Cr_2_O_7_^2−^, HCrO_4_^−^ and CrO_4_^2−^ [12,41]. At the same time, the surface of La-loaded biochar is positively charged; therefore, under acidic conditions, the adsorption mechanism of chromate by La-BC may be explained as an electrostatic interaction [14,32]. The change of pH before and after the adsorption proves another possible mechanism: the final pH of the solution was higher than the initial pH, indicating that OH^−^ was released during the process of the chromate adsorption, which confirms that a ligand exchange interaction was involved [30]. Based on the above, several possible adsorption equations are summarized as follows:La_2_O_3_ + HCrO_4_^−^ + H^+^ → La_2_CrO_6_ + H_2_O(1)
La_2_O_3_ + Cr_2_O_7_^2−^ + H^+^ → La_2_Cr_2_O_9_ + OH^−^(2)
La_2_O_3_ + CrO_4_^2−^ + H_2_O → La_2_CrO_6_ + 2OH^−^(3)

With the increase of pH (>pH_pzc_), a large amount of OH^−^ was released into the solution, which can compete with chromate ions for active adsorption sites, and can generate electrostatic repulsion [23]. All of the above resulted in a significant decrease of the adsorption of the chromate at higher pH values; therefore, the range of pH > 7 was no longer taken into account in the subsequent experiments.

The XRD images of La-BC (900 °C) before and after the adsorption of the dichromate solutions with a pH range of 2–7 were compared (Figure 7b). The results showed that when pH > 2, the main peaks after adsorption were characteristic peaks of La (OH)_3_ (PDF#36-1481). When pH = 2, the XRD image of the La-BC was a smooth curve without any peak. The reason may be that the La_2_O_3_ in the La-BC (900 °C) was largely dissolved into the solution in a strong acidic environment. For this reason, the amount of La dissolution of the adsorbed dichromate solution was detected (Appendix A), and the results confirmed that a large amount of La was dissolved at pH = 2. The initial pH of the dichromate solution was about 4–5; within this range, not only can the adsorption amount of La-BC be guaranteed not to be too low but also the dissolution amount of La can be kept at a low level. Based on the results, it was finally decided that the experiment would not take too much adjustment to the pH of the initial solution, and it should be kept at pH = 4.6.

### 3.7. Effect of Coexisting Acid Ions on the Adsorption Experiments

Five kinds of 100 mg/L K_2_Cr_2_O_7_ solutions containing NaCl, Na_2_SO_4_, NaNO_3_, NaHCO_3_ and Na_2_HPO_4_ were prepared, in which the concentrations of the coexisting ions were 0.05 mol/L, 0.1 mol/L and 0.2 mol/L, respectively. The pure 100 mg/L K_2_Cr_2_O_7_ was used as a blank control, and the adsorption experiments were carried out under the conditions of 300 rpm, 303 K, contact time 5 h, pH 4.6 and a 1 g/L dosage of La-BC (900 °C). It was shown that different interfering ions had different influences on the adsorption amount of Cr (VI) (Figure 8a). The results showed that Cl^−^ and NO_3_^−^ had no effect on the adsorption of chromate, and SO_4_^2−^ had little effect. However, HCO_3_^−^ and HPO_4_^2−^ seriously interfered the adsorption of chromate, which may be for the following two reasons: (1) the structure of these oxygen-containing acid groups was similar to that of Cr_2_O_7_^2−^ and CrO_4_^2−^, resulting in competition in the adsorption process; (2) both carbonic acid and phosphoric acid are weak acids, so the hydrolysis degree of HCO_3_^−^ and HPO_4_^2−^ in the solution is greater than the degree of ionization—this is why NaHCO_3_ and Na_2_HPO_4_ solutions are weakly alkaline. The release of OH^−^ led to the decrease of the adsorption amount of the chromate.

### 3.8. Cycle Experiments

The experiment of the adsorption of Cr (VI) by La-BC (900 °C) was carried out under the conditions of an initial Cr (VI) concentration 200 mg/L, pH 4.6, 300 rpm, 303 K, contact time 5 h and a 1 g/L adsorbent dosage, and five cycles were carried out for the desorption treatment of 1 mol/L NaOH (Figure 8b). After each cycle, the recovered product was calcined in inert gas at a high temperature in order to ensure that the composition of the recovered product was the same as the original La-BC material. The experimental results showed that La-BC (900 °C) had good stability, a low loss in five cycles, and a good reuse effect. The circulating experiment also proved the possibility of the use of La-BC in treating chromium-containing wastewater.

### 3.9. Adsorption Mechanism

The pH experiments and XRD characterization results indicated that the adsorption behavior of La-BC on Cr (VI) might have an electrostatic interaction and a ligand exchange (Figure 7). In addition, X-Ray photoelectron spectroscopy analysis was performed on the La-BC (900 °C) after the adsorption of chromite, and a comparison was made with an XPS image of La-BC (900 °C) before the adsorption (Figure 9). The wide-scan XPS spectrum of La-BC after the experiments showed that the peak of the Cr element appeared in La-BC (900 °C) (Figure 9a). Moreover, the corresponding peaks where the binding capacity was 577.8 eV, 579.7 eV proved that Cr (VI) was successfully absorbed (Figure 9b) [42]. The Cr has been absorbed successfully. In the La3d images, the binding energy of the element La had a negative displacement (0.4–0.7 eV) after adsorption (Figure 9c), indicating that electron transfer occurred between La-BC (900 °C) and chromite, which formed a strong chemical bond [30]. The O1s spectrum changed significantly before and after the adsorption, which could explain the adsorption mechanism (Figure 9d,e). The O1s deconvolution (Figure 9d) indicated that three main oxygen components—lattice oxygen La-O (530.8 eV), OH^−^ (531.9 eV), and adsorbed H_2_O (533.0 eV)—existed in the La-BC. However, after the Cr (VI) sorption, the peak of La-O nearly disappeared, while the H_2_O percentage increased (Figure 9e). This was because La_2_O_3_ is hygroscopic, and it adsorbed moisture to become La (OH)_3_ [28]. All of the above results show that the adsorption may be caused by ligand exchange or complexation. The increase of the proportion of the element O from 28.96% to 32.47% might also be explained by the adsorption of the oxygen-containing acid radicals CrO_4_^2−^ and Cr_2_O_7_^2−^.

## 4. Conclusions

In this experiment, wet spinning and ion exchange were combined to load La^3+^ onto sodium alginate fiber, and then La^3+^ was converted to La_2_O_3_ in the process of high-temperature carbonization. Lanthanum-oxide–modified biochar showed an excellent adsorption capacity for chromate, and it also had a high adsorption capacity in a potassium dichromate solution with a high concentration at room temperature, and its saturated adsorption capacity was about 104.9 mg/g. The isotherm data showed that the adsorption process is more consistent with the Langmuir model, and that it has a higher correlation with the adsorption of a single molecular layer. The adsorption kinetic data followed the pseudo-second-order kinetic model, showing that the adsorption process was dominated by chemisorption. The thermodynamic parameters proved that the adsorption of chromate by La-BC is spontaneous and endothermic. In addition to physical adsorption, the adsorption mechanism of Cr (VI) in La-BC may be electrostatic interaction, ligand exchange, or complexation (Figure 10). Compared with Cr (VI) adsorbents in other literatures (Table 1), La-BC has the advantages of a large adsorption capacity, a simple preparation process and good recycling performance.

Therefore, La-BC can be used as a new type of environmentally friendly adsorbent, which has great research significance and broad development prospects in the field of sewage treatment.

## Figures and Tables

**Figure 1 materials-14-02224-f001:**
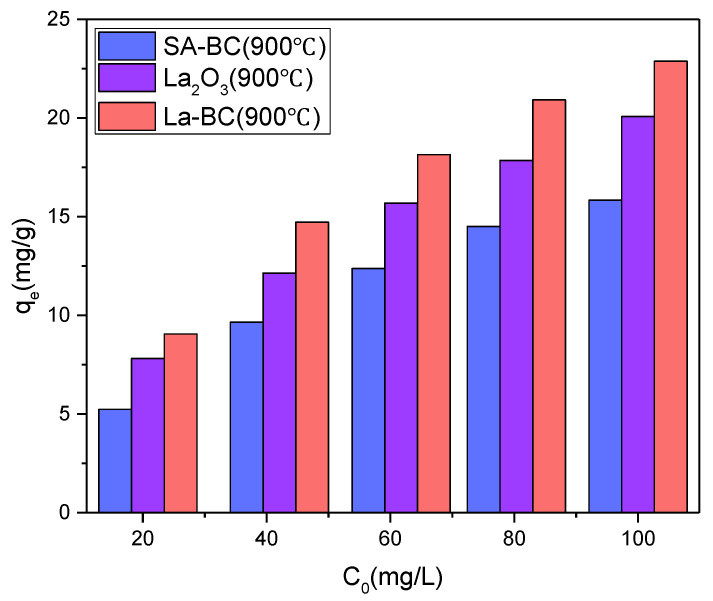
Pre-experiment: comparison of the adsorption capacity of Cr (VI) by SA-BC, La_2_O_3_ and La-BC.

**Figure 2 materials-14-02224-f002:**
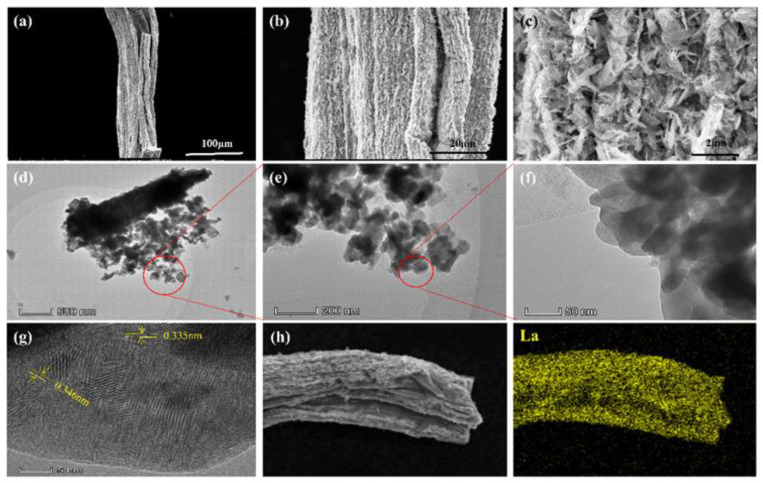
(**a**–**c**) SEM images of La-BC; (**d**–**g**) TEM images of La-BC; (**h**) elemental mapping images of La-BC.

**Figure 3 materials-14-02224-f003:**
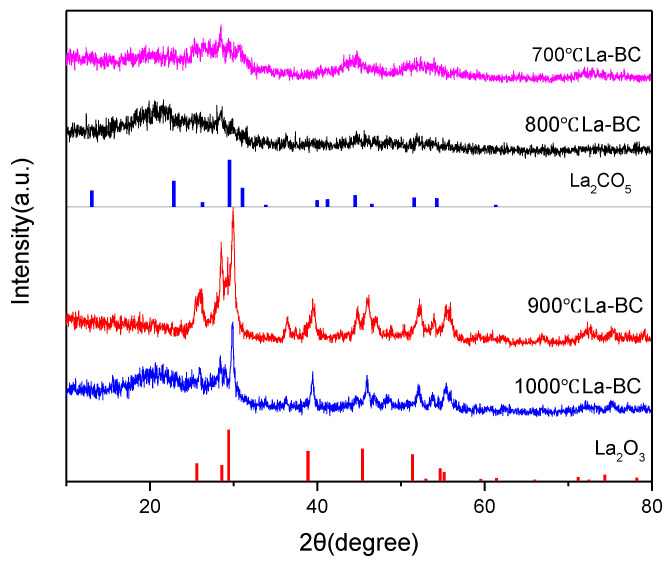
XRD patterns of the different carbonization temperatures of the La-BC samples.

**Figure 4 materials-14-02224-f004:**
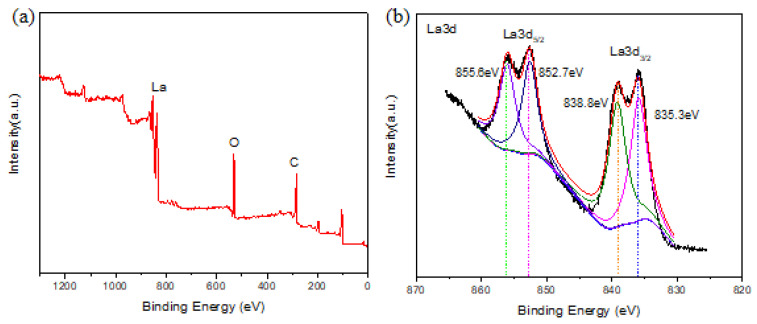
(**a**) XPS spectra of the La-BC (900 °C) before Cr (VI) adsorption; (**b**) La 3d spectra of La-BC (900 °C) before Cr (VI) adsorption.

**Figure 5 materials-14-02224-f005:**
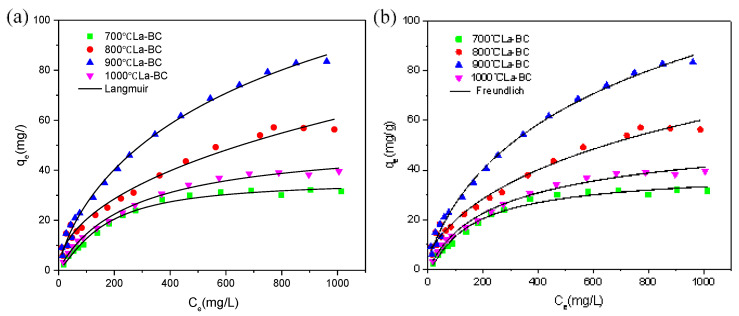
Fitting the adsorption data of Cr (VI) on La-BC (900 °C) to (**a**) the Langmuir isotherm, and (**b**) the Freundlich isotherm.

**Figure 6 materials-14-02224-f006:**
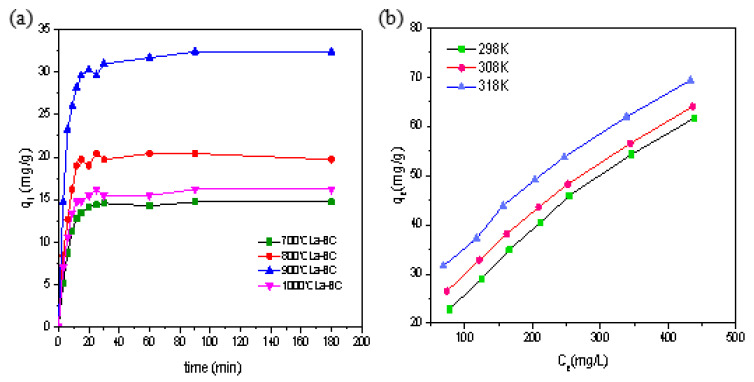
(**a**) Relationship between the adsorption amount of Cr (VI) and the adsorption time by La-BC at four carbonization temperatures; (**b**) relationship between the adsorption amount of Cr (VI) and the adsorption temperature by La-BC (900 °C).

**Figure 7 materials-14-02224-f007:**
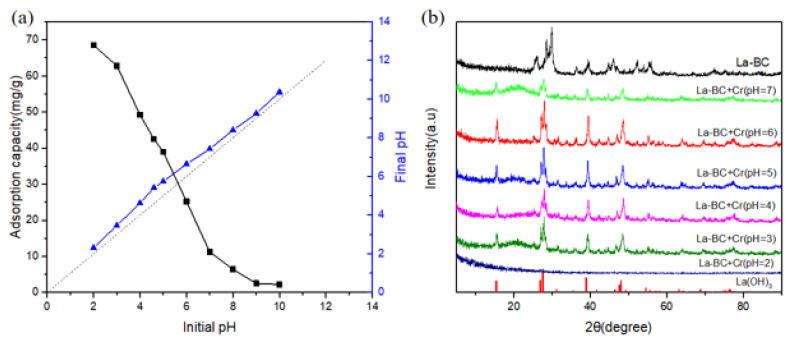
(**a**) The effect of pH on the adsorption amount, and the comparison of the initial and final pH; (**b**) the XRD images of La-BC (900 °C) after the adsorption with a pH range of 2–7.

**Figure 8 materials-14-02224-f008:**
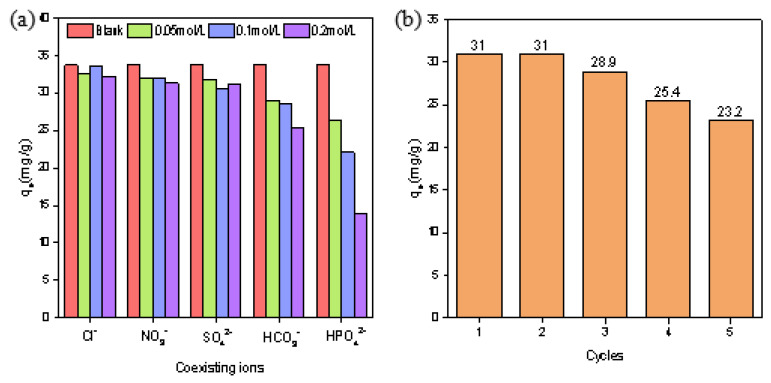
(**a**) Effect of coexisting ions on the adsorption amount of Cr (VI); (**b**) five cycles of Cr (VI) adsorption by La-BC (900 °C).

**Figure 9 materials-14-02224-f009:**
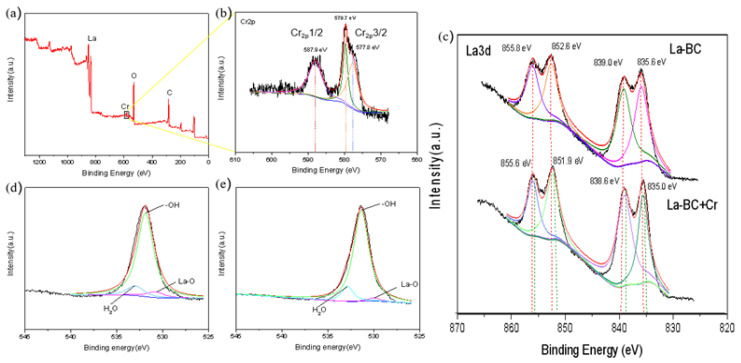
(**a**) XPS spectra of the La-BC (900 °C) after Cr (VI) adsorption(a); (**b**) Cr2p; (**c**) La3d; (**d**) the O1s spectra of La-BC (900 °C) before and (**e**) after Cr (VI) adsorption.

**Figure 10 materials-14-02224-f010:**
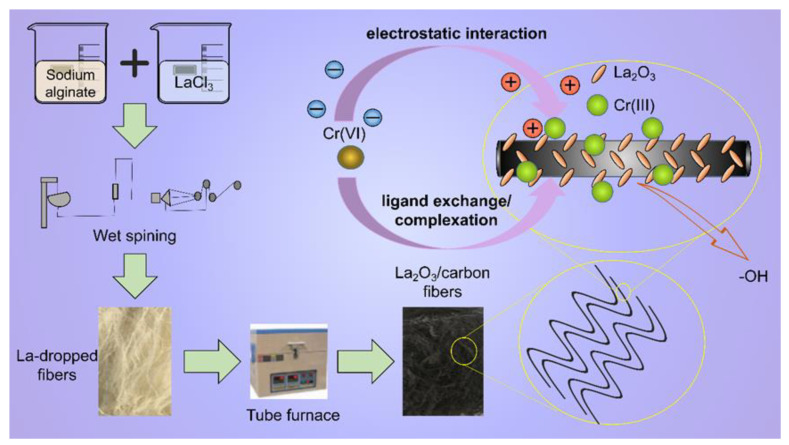
Preparation process of La-BC and the mechanism of Cr (VI) adsorption with it.

**Table 1 materials-14-02224-t001:** Literature data on the adsorption of Cr (VI) ions by various adsorbents.

Adsorbent	Adsorption Capacity (mg/g)	Ref.
MgAl-LDH	84.2	[42]
MWCNTs-70	17.2	[15]
Chitosan engraved iron and lanthanum Mixed oxyhydroxide	106.04	[32]
Fe_3_O_4_ nanoparticles functionalized polyvinyl alcohol/chitosan magnetic composite hydrogel	24.69	[43]
Chitosan-citric acid nanoparticles	106.15	[44]
Cross-linked chitosan bentonite composite	89.13	[45]
Fe–Mn oxide-modified biochar	59.8	[16]
Biochar supported nZVI composite	31.53	[1]
La_2_O_3_-dropped biochar	104.9	This article

## Data Availability

Data sharing is not applicable for this article.

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
