# Peer review of "Adsorption of Hexavalent Chromium by Sodium Alginate Fiber Biochar Loaded with Lanthanum"

_materials, 2021, doi:10.3390/ma14092224_

Round 1
Reviewer 1 Report
Dear Prof. Yuqian Cui, your manuscript entitled "Adsorption of hexavalent chromium by sodium alginate fiber biochar loaded with lanthanum" by Xinzhe Sun, Yuanyuan Sun and Yuqian Cui, has been written, developed and presented. In addition, it is of great interest in this area of research. The isotherm data are in good sagreement with Langmuir model, showing a
single molecular layer. The adsorption kinetic data and the thermodynamic parameters are well described. Moreover, you described very well that the adsorption mechanism of Cr(VI) in La-BC may be electrostatic interaction, ligand exchange or complexation. This research may be of great technological interest.
However, from the point of view of the structural characterization I believe that the interpretation of the x-ray diffraction diagrams (figure 2) should be done in more detail, to ensure that the studied materials undoubtedly correspond to the given compositions. You should index these diagrams in detail, especially for the 900 and 1000 ºC phases.
Reviewer 2 Report
Manuscript Number: materials-1067537
Title: Adsorption of hexavalent chromium by sodium alginate fiber biochar loaded with lanthanum
The present manuscript discussing the synthesis of lanthanum incorporated alginate based biochar for the removal of Cr(VI). Thus prepared materials were characterized by different physicochemical analysis. The objective of the manuscript is not clear in the current manuscript. There are several lanthanum based adsorbents are available in the literature for the removal of Cr. But, there is no information regarding the choice of La2O3 incorporated biochar for the removal of Cr is available in the current manuscript. The effect of biochar on the adsorption of Cr should be detailed in the manuscript. Moreover, the existing literature regarding the La based sorbents for the removal of Cr should be incorporated and how these study superior to the existing literature should be emphasized in the introduction section. Based on these I am reluctant to recommend this article for the publication. More detailed comments are appended below,
- First of all, why these La incorporated in the biochar. What is the effect of Biochar on the adsorption of Cr. In general the doping of La would enhance the dispersibility of La, thereby increase the La utilization efficiency (ref. Chemical Engineering Journal 317 (2017) 1059–1068). This also enhance the kinetics of adsorption. To validate this the adsorption experiments of pure biochar and bulk La2O3 synthesized under identical condition of composite should be studied. It also provide the clue that how these bio char assisting the adsorption of Cr. These details should be included without fail.
- Line 73, Please provide the references.
- Section 2.1, the purity of all the samples should be included.
- Section 2.3, the experimental condition should be included.
- Section 2.4, the exact details of adsorption such as concentration, solid-to-liquid ratio should be provided for all the parameters for the reproduction of the experiments. The concentration of Cr for kinetic experiments are missing in the manuscript.
- Line 156, La2CO5???.
- Line 184, the adsorption values should be rounded to one decimal points. Please check throughout the manuscript.
- Figure 5b, Replot the figure with equilibrium concentration.
- Section 3.5, it is known that the at lower pH the Cr exist as Cr2O72- and at higer pH the dichromate dissociate in to two CrO42-. It increase the concentration of Cr species. It may results the decrease in the adsorption capacity. How author tackled this situation. The details should be included. The dichromate is used mostly to study the adsorption at lower pH only. Please comment on this.
- Figure 6, is the adsorption of Cr at pH 4.6 included in the figure. Please provide the leaching of La3+ in the figure.
- Line 316, What is the source of Cr3+, is it reduced during adsorption, if so necessary details should be provided (Ref. ChemistrySelect 2017, 2, 10459 –10469).
- Section 3.7, Is these materials after 1 cycle is calcined or used without calcination. The use of NaOH would convert the La-CrO4 in to La-OH. These materials different from La2O3. How one can expect same adsorption mechanism from different materials. Please comment on this.
Reviewer 3 Report
The MS by X. Sun, Y. Sun and Y. Cui concerning with the use of La2O3 modified biochar for chromates adsorption. This research has a strong environmental focus aiming adsorbents with high efficiency. In this way authors demonstrated that lanthanum oxide modified biochar showed excellent adsorption capacity for chromate, and it also had a high adsorption capacity in potassium dichromate solution with high concentration at room temperature, and its saturated adsorption capacity was about 104.93 mg/g. Authors provide a high quality thermodynamic and kinetic data concerning with adsorption process. The MS can be published in Materials after taking into account these points:
1) The quality of some Figures should be improved (like Fig 2 and so on);
2) Fig. 2: Powder diffraction pattern 900 C La-BC contains an additional reflections caused by appearing of the additional phase. Please, provide the information about this phase.
3) Please add some discussion about comparison of your sorbent with literature reported analogs. This data will increase the strength of the MS. Moreover some comparison comments should be added to the Conclusions section.
Reviewer 4 Report
Manuscript: Adsorption of hexavalent chromium by sodium alginate fiber 2 biochar loaded with lanthanum
Comments. Authors report a study on the adsorption of Cr(VI) by La2O3 modified biochar fiber. Study is well conducted and different aspects concerning the adsorption process are evaluated: isotherms, kinetics, thermodynamics, together with the study of the effect of some experimental conditions such as pH, temperature, coexistence of ions.
I think manuscript is well structured and complete and, therefore, suitable for publication on Materials without further revisions.
Round 2
Reviewer 2 Report
Manuscript Number: materials-1067537-v2
Title: Adsorption of hexavalent chromium by sodium alginate fiber biochar loaded with lanthanum
I am personally not satisfied with the response made to the comments 1, 4 and 9. Authors are very adamant to make changes based on these comments as these are very important query for scientific article. Author should realize that the comments are to improve the scientific content of the manuscript. Without these data, the manuscript cannot be accepted for publication.
Comment 1: First of all, why these La incorporated in the biochar. What is the effect of Biochar on the adsorption of Cr. These details should be included. In science there is always possibility to validate the assumption. For author’s kind information. The La2O3 could be prepared by calcining LaCl3 salt at same calcination procedure. Similarly the biochar could be prepared without using LaCl3 or by using inert NaCl as coagulant. It may provide the specific individual materials. In this manuscript, there is no details about the sorption efficiency of biochar or La. It provides only the La-Biochar Cr adsorption efficiency. In science always the curiosity arises whether the adsorption takes place by La or biochar is remain mistry in the current manuscript.
Comment 4: The experimental condition should be provided. Author should realize it is scientific article. The experimental may be validated by different researcher in this field. The analysis feature is changing with experimental condition. Please ask your sample analyzer to provide these details they may be very happy to send you these details. Be a researcher instead of just data communicator.
Comment 9: Regarding the speciation of Cr. Author are encouraged to study the Cr speciation. The response seems the author’s inability to understand the science. The dissociation of Cr2O72- (one species) increase the Cr species in to two (2CrO42-). Done make such foolish response. Hope author understand the query.
There are several lanthanum based adsorbents are available in the literature for the removal of Cr. But, there is no information regarding the choice of La2O3 incorporated biochar for the removal of Cr is available in the current manuscript. The effect of biochar on the adsorption of Cr should be detailed in the manuscript. Moreover, the existing literature regarding the La based sorbents for the removal of Cr should be incorporated and how these study superior to the existing literature should be emphasized in the introduction section.
Round 3
Reviewer 2 Report
Authors are very adamant to make changes in the manuscript and their responses are very unethical. None of the corrections were made for the comments1, 9.